

# Ozonolysis of α-phellandrene – Part 2: Compositional analysis of secondary organic aerosol highlights the role of stabilised Criegee intermediates

Felix A. Mackenzie-Rae[1], Helen J. Wallis[2], Andrew R. Rickard[2,3], Kelly Pereira[2], Sandra M. Saunders[1], Xinming Wang[4,5], Jacqueline F. Hamilton[2]

[1]School of Molecular Sciences, The University of Western Australia, Crawley WA 6009, Australia
[2]Wolfson Atmospheric Chemistry Laboratories, Department of Chemistry, University of York, York, YO10 5DD, UK
[3]National Centre for Atmospheric Science, University of York, York, YO10 5DD, UK
[4]State Key Laboratory of Organic Geochemistry and Guangdong Key Laboratory of Environmental Protection and Resources Utilization, Guangzhou Institute of Geochemistry, Chinese Academy of Sciences, Guangzhou 510640, China
[5]Center for Excellence in Regional Atmospheric Environment, Institute of Urban Environment, Chinese Academy of Sciences, Xiamen 361021, China

*Correspondence to*: J. Hamilton (jacqui.hamilton@york.ac.uk)

**Abstract.**

The molecular composition of secondary organic aerosol (SOA) generated from the ozonolysis of α-phellandrene is investigated for the first time using high pressure liquid chromatography coupled to high-resolution Quadrupole-Orbitrap tandem mass spectrometry. In total, 21 prominent products or isomeric product groups were identified using both positive and negative ionisation modes, with potential formation mechanisms discussed. The aerosol was found to be composed primarily of polyfunctional first- and second-generation species containing one or more carbonyl, acid, alcohol and hydroperoxide functionalities, with the products significantly more complex than those proposed from basic gas-phase chemistry in the companion paper (Mackenzie-Rae et al., 2017a). Mass spectra show a large number of dimeric products are also formed. Both direct scavenging evidence using formic acid, and indirect evidence from double bond equivalency factors, suggests the dominant oligomerisation mechanism is the bimolecular reaction of stabilised Criegee intermediates (SCIs) with non-radical ozonolysis products. Saturation vapour concentration estimates suggest monomeric species cannot explain the rapid nucleation burst of fresh aerosol observed in chamber experiments, hence dimeric species are believed to be responsible for new particle formation, with detected first- and second-generation products driving further particle growth in the system. Ultimately, identification of the major constituents and formation pathways of α-phellandrene SOA leads to a greater understanding of the atmospheric processes and implications of monoterpene emissions and SCIs, especially around Eucalypt forests regions where α-phellandrene is primarily emitted.

## 1 Introduction

Aerosols are abundant in the atmosphere, playing an important role in the climate system by scattering and absorbing



radiation, influencing cloud formation and participating in heterogeneous chemical processes (Carslaw et al., 2010; Hallquist et al., 2009; IPCC, 2013). Furthermore, it is well established that atmospheric aerosols have an important impact on local air quality and human health, with exposure associated with detrimental effects on the respiratory and cardiovascular systems (Brook et al., 2010; Pope and Dockery, 2006). Nevertheless, there remains a large amount of uncertainty regarding the true

impact of atmospheric aerosols on climate and health, driven by a general lack of understanding in sources, formation mechanisms, composition and properties (Hallquist et al., 2009).

A common feature of atmospheric aerosol is the presence of condensed secondary organic material, formed as a result of oxidation of volatile organic compounds (VOCs) into less volatile, condensable, species (Jimenez et al., 2009; Kanakidou et

al., 2005). With significant biogenic emissions (Guenther et al., 1995, 2012), high chemical reactivity (Atkinson and Arey, 2003) and large secondary organic aerosol (SOA) formation yields (Griffin et al., 1999; Hoffmann et al., 1997), monoterpenes are important contributors to organic aerosol globally, with their ozonolysis a dominant pathway for SOA formation and growth (Jokinen et al., 2015; Ortega et al., 2012; Zhao et al., 2015a). One monoterpene for which little study has been conducted is α-phellandrene. With an ozonolysis rate constant of $3.0 \times 10^{-15}$ ($\pm$ 35%) $cm^3$ molecule$^{-1}$ s$^{-1}$, α-

phellandrene is one of the most reactive monoterpenes (Atkinson and Arey, 2003). α-Phellandrene has been measured in field studies in Brazil (Kesselmeier et al., 2000) and Japan (Ramasamy et al., 2016), although it has been noted that ambient detection is made difficult by its extremely high reactivity (Geron et al., 2000; Saxton et al., 2007). Believed then to be of particular importance is the abundance of α-phellandrene in extracts of numerous species of Eucalypts (Brophy and Southwell, 2002; Li et al., 1995; Maghsoodlou et al., 2015; Pavlova et al., 2015), with Malekina et al. (2009) and He et al.

(2000) having identified α-phellandrene in emissions of various Eucalypt species in the laboratory. Native to Australasia, the remarkable adaptability, rapid growth rates and high quality wood of Eucalypts has led to plantation forestry in more than 100 countries spanning six continents (> 20 million ha), making Eucalypts the most widely planted hardwood forest trees in the world (Myburg et al., 2014).

In the companion paper (Part 1, Mackenzie-Rae et al., 2017a), it was found that α-phellandrene yields a large amount of self-nucleated SOA upon reaction with ozone, concluding that the ozonolysis of α-phellandrene is likely an important contributor to the intense and frequent nocturnal nucleation events observed in Eucalypt forests (Lee et al., 2008; Ortega et al., 2012; Suni et al., 2008). However, postulated gas-phase species could not explain the properties of the SOA observed. Nucleation was inhibited in experiments that introduced a Criegee intermediate (CI) scavenger into the system, thereby identifying CIs

as important precursors to nucleating compounds, whilst temporal growth profiles showed both first- and second-generation products to be major contributors to the SOA mass. This finding is consistent with recent literature that is building around stabilised Criegee intermediates (SCIs) as important precursors to SOA nucleation and growth, although the mechanism describing how SCIs form condensable products is still open to debate (Bateman et al., 2009; Bonn et al., 2002; Kristensen et al., 2016; Sadezky et al., 2006, 2008; Sakamoto et al., 2013, Ahmad et al., 2017).



This paper extends the current discussion by analysing filter samples collected during the α-phellandrene ozonolysis chamber experiments (as described in Mackenzie-Rae et al., 2017a). Samples were analysed using a combination of liquid chromatography with soft-ionisation electrospray mass spectrometry. This technique is sensitive towards polar organic molecules containing basic or acidic (ionisable) functional groups, with the technique having previous success in identifying compounds within SOA generated from monoterpene ozonolysis (e.g. Bateman et al., 2009; Camredon et al., 2010; Walser et al., 2008; Witkowski and Gierczak, 2017). The objective of this study is therefore to identify SOA constituents from the gas-phase ozonolysis of α-phellandrene, and their associated formation mechanism for the first time.

## 2 Method

### 2.1 Chamber experiments

SOA was generated and sampled during eleven dark α-phellandrene ozonolysis experiments using the indoor smog chamber facility at the Guangzhou Institute of Geochemistry, Chinese Academy of Sciences (GIG-CAS, Wang et al., 2014). The complete experimental design and details are provided in the companion paper (Mackenzie-Rae et al., 2017a). Briefly and relevant to the current discussion, α-phellandrene (10 – 175 ppb, Aldrich Chemical Company, Inc., USA) and ozone (56 – 500 ppb) were mixed in a background matrix of purified air. All experiments were conducted under low humidity conditions (RH ≤ 5%), room temperature (~298 K) and ambient pressure (~760 Torr). All but two experiments (9 and 11) had anhydrous cyclohexane (Sigma-Aldrich, 99.5%) added in sufficient quantity as an OH radical scavenger, with one of the non OH-scavenged experiments having 385 ppb of $NO_2$ added (experiment 11). Two experiments (6 and 7) had 800 ± 80 ppb formic acid (J&K Scientific Ltd., 98%) added as a SCI scavenger (Bonn et al., 2002; Winterhalter et al., 2009). Particle number size distributions were measured by a scanning mobility particle sizer (SMPS; TSI Incorporated, USA) (Wang and Flagan, 1990), with particle mass concentrations calculated to range from 21 – 660 µg m$^{-3}$, whilst particle chemical composition was measured in real-time by a high-resolution time-of-flight aerosol mass spectrometer (AMS; Aerodyne Research Incorporated, USA) (DeCarlo et al., 2006; Jayne et al., 2000). Aerosol samples were collected on pre-fired Whatman quartz microfiber filters (47 mm) by pumping air through the filter at 16 – 25 slpm, for a minimum of two hours after chamber conditions had reached a steady state. Filter samples were then wrapped in foil and stored at 4°C prior to analysis. Specific starting conditions and information pertaining to SOA collection for each of the 11 experiments are presented in the Table 1.

### 2.2 Sample preparation and analysis

Filter samples were extracted by sonication using LC-MS Optima grade water (Fisher Scientific) as the solvent (Hamilton et al., 2008). The sample solution was then filtered through a 0.45 µm pore syringe (Millex-HA), with a further 1 mL of water added to the dry filter paper residue for a second extraction following the same method. Combined extraction samples were



then evaporated to dryness using a V10 vacuum solvent evaporator (Biotage, 36°C, 8 mbar). Lastly, the residue was re-suspended in 500 µL of 50:50 methanol:water (Optima LC-MS grade, Fisher Scientific) solution, with dissolution aided by the Biotage evaporator (4000 rpm). Sample exposure to ambient light was minimised throughout the extraction process. Pre-conditioned filters, prepared by heating blank filter papers (Whatman Quartz microfiber filters, 47 mm) in a furnace at 550°C

for 5 hours, were analysed as a procedural blank following the same method.

Samples were analysed using reverse-phase high-performance liquid chromatography (HPLC) (Agilent 1100 series, Berkshire, UK) coupled to a Q-Exactive Hybrid Quadrupole-Orbitrap Mass Spectrometer (Thermo Scientific). Samples (30 mL) were injected using an autosampler and separated using a Pinnacle DB $C_{18}$ column with 5 µm particle size (Thames

Restek, UK). The starting mobile phase consisted of a gradient from 85% water with 0.1% formic acid (Sigma Aldrich), and 15% methanol (LC-MS Optima grade, Fisher Scientific), to 10% of the water/formic acid solution and 90% methanol over 13 minutes, before holding this ratio for a further minute. Electrospray ionisation (ESI, 35 eV) was used in both positive and negative modes with full mass spectra scans acquired between $m/z$ 50 – 750 range with a resolution of 70,000 at $m/z$ 200. Ions were additionally fragmented using high-energy collisional dissociation to yield tandem mass spectrometry (MS/MS)

for structural identification. The MS was frequently mass-calibrated using ESI positive and negative ion calibration solutions (Pierce, Thermo Scientific). Data analysis were conducted using Xcalibur mass spectrometry software, assuming the only atoms present were $^{12}$C, $^{13}$C, $^{1}$H and $^{16}$O in the negative ionisation mode in addition to $^{23}$Na in the positive ionisation mode. Analysis of the pre-conditioned filters showed only baseline noise, confirming that the peaks in the chromatograms were not due to artefacts of sample preparation. No evidence of methylation, which has been reported in other works using methanol

as the extracting solvent (Walser et al., 2008), was observed in any of the tandem mass spectra.

## 3 Results and discussion

### 3.1 Product identification

Because all first- and second-generation products of α-phellandrene ozonolysis contain at least one functional group that is capable of ionisation (Mackenzie-Rae et al., 2016, 2017a), it is reasonable to assume that a high proportion of water soluble

SOA components will be observed, with the analyte R being observed as $[R–H]^-$ ion in the negative mode and $[R+Na]^+$ ion in the positive mode. Negative mode analysis leads to formation of deprotonated ions; hence molecules containing functional groups that readily lose a proton, such as carboxylic acids, are frequently observed in this mode. Meanwhile in positive mode analysis, ions are produced by protonation or cationisation with sodium ions ($Na^+$), which are present in the glassware used and, in the case of oxygenated functionalities, can improve detection and sensitivity (Hamilton et al., 2008). Hence groups

that more readily accept a positive charge, such as carbonyls, are often observed in positive mode analysis, which are expected to be dominated by sodium ion adducts (Camredon et al., 2010). Accurate mass measurements were used to assign molecular formulae to SOA components with mass accuracy of less than 2 ppm. α-phellandrene has an interesting structure



for understanding SOA formation mechanisms. It has two double bonds within the ring; reaction with ozone at the first bond opens the ring, while the second ozone addition causes fragmentation. Thus the dominant SOA components from the first oxidation will likely have 10 carbons. Those produced from the second will have seven or three carbons, with the latter unlikely to form significant SOA mass. The following sections will focus initially on the composition of SOA formed during standard α-phellandrene ozonolysis experiments. The final section will focus on the impact of adding a SCI scavenger during the experiments.

### 3.1.1 Negative mode

The chromatograms produced using negative mode ionisation were reasonably consistent across the experimental suite, showing the same characteristic major peaks irrespective of experimental conditions. Two example chromatograms, from experiments with and without formic acid, are shown in Fig. 1 with major peaks detected across all experiments labelled in retention time order. Using high mass accuracy tandem mass spectra, the structure of 12 major compounds have been identified (Table 2), with structural elucidation consistent with postulated degradation mechanisms (Mackenzie-Rae et al., 2016, 2017a). Labelled tandem mass spectra are provided in the Supplement (Sect. S1). All identified species contain carboxylic acid functionality, with a characteristic loss of 44 Da from the parent signal observed in all tandem mass spectra, showing a clear bias in negative mode ESI towards compounds containing acid functionality.

The peak at retention time 7.06 minutes (N8) is the major peak in all negative mode chromatograms, with $[M–H]^- = m/z$ 159 and a neutral formula of $C_7H_{12}O_4$. A molecular formula with 7 carbon atoms indicates a second-generation product. In addition to a loss of 44 Da, the tandem mass spectra is also characterised by a loss of 18 Da, suggesting the presence of further functionalisation, in this case likely an alcohol group. Two plausible isomeric species have been suggested based on this information (Table 2). As shown in Fig. 2 for one isomer, acid formation occurs through stabilisation of the CI from one ozone addition, whilst the CI from the other addition decomposes via the hydroperoxide channel; whereby the excited CI isomerises through a H-shift to form a vinylhydroperoxide which subsequently decomposes into a vinoxy radical and a hydroxyl radical (Johnson and Marston, 2008; Niki et al., 1987). Molecular oxygen then adds to the vinoxy radical which ultimately yields an alcohol group. Both of these are major reaction pathways and can explain the high abundance of compound N8 in α-phellandrene SOA. Theoretical results show the energy barrier for accessing the hydroperoxide channel from the primary CI generated from the more substituted double bond is around 20 kJ mol$^{-1}$ lower in energy than the respective barrier for accessing the hydroperoxide channel from the relevant CI from the less substituted double bond (Mackenzie-Rae et al., 2016). Based on this information, the secondary-alcohol isomer (formation mechanism shown in Fig. 2) is favoured as compound N8.

Alternatively, if the peroxy radical formed during the hydroperoxide channel were to form a ketone rather than an alcohol, then compound N1 would be formed, as shown in Fig. 2. This product was identified at retention time 3.81 minutes, with a





[M–H]⁻ = $m/z$ 157 and a neutral formula of $C_7H_{10}O_4$. The mechanism yielding adjacent aldehyde and ketone functional groups is only accessible for the CIs formed from ozonolysis of the more substituted double bond in α-phellandrene, with the other CIs unable to form a ketone without fragmenting, due to the α-carbons being tertiary. This comparative lack in mechanistic flexibility, along with a higher vapour pressure, may explain why compound N1 is relatively minor compared to
N8.

The prominent peaks at retention times 5.01 (N4) and 5.39 minutes (N5) are due to isomeric molecules with [M–H]⁻ = $m/z$ 175 and a neutral formula of $C_7H_{12}O_5$. These two peaks are observed in all samples analysed, with varying relative intensities. Tandem mass spectra for these two species are similar with a number of common ion peaks, suggesting similar
structures containing acid and hydroxyl functionalities, differing only by positioning of these functional groups within the molecule. Nonetheless, a small number of minor differences in MS/MS fragmentation enables structural elucidation. N4 is formed through initial ozonolysis of the least substituted double bond, with the second addition of ozone participating in the hydroperoxide channel on the fragment $C_7$ backbone, before undergoing a number of isomerisations, as shown in Fig. 3. Meanwhile N5 is formed from ozone addition occurring in the reverse order. In general, the peak area of N4 is around twice
as large as that for N5. The exact structures were confirmed by comparison to reference spectra available in the online mzcloud database (https://www.mzcloud.org/), with similarities of 94 and 96 % for N4 and N5 respectively (see Supplement Sect. S1). Given vapour pressures are likely to be very similar (Table 2), this observation suggests that either initial attack at the least substituted double bond in a-phellandrene is favoured, or the CI formed from ozonolysis of the more substituted double bond in α-phellandrene is more likely to participate in the hydroperoxide channel than the CI from ozonolysis of the
less substituted double bond, both of which are consistent with recent theoretical findings (Mackenzie-Rae et al., 2016). Relative to compound N8, peak areas for N4 and N5 are suppressed upon introduction of $NO_2$ into the system (experiment 11, also no cyclohexane). Similar inhibition is not observed in the other ozonolysis experiment where cyclohexane was not added (experiment 9); thus it is concluded that it is the addition of $NO_2$ to the system that precludes formation of compounds N4 and N5 by preferentially reacting with the acyl peroxy radical. Therefore it is expected that compounds N4 and N5
become increasingly significant contributors to SOA generated from α-phellandrene under increasingly 'NOₓ-limited' conditions.

Peaks at retention times 4.15 (N2) and 4.86 (N3) minutes are found consistently among the chromatograms from all samples, and correspond to products with [M–H]⁻ = $m/z$ 145 and 187 and neutral formulas of $C_6H_{10}O_4$ and $C_7H_8O_6$ respectively. The
high oxygen content of both products indicates that at least one of the ozone additions decomposes through the hydroperoxide channel, with further intramolecular H-transfers and oxygen additions possible throughout radical decomposition (Crounse et al., 2013; Kurtén et al., 2015). Proposed structures for compounds N2 and N3, derived from tandem mass spectra (Supplement Sect. S1), are given in Table 2.



Compound N6 and N7 at retention times 6.01 and 6.23 minutes, have a [M–H]⁻ = $m/z$ 143 and are assigned a neutral formula of $C_7H_{12}O_3$. These species are formed through the CIs from both ozone additions being stabilised, resulting in aldehyde and carboxylic acid functionality as shown in Fig. 2 for N6. The impact of a SCI scavenger on the relative abundance of these species is discussed further in section 3.5.

Observed in the majority of chromatograms, compound N11 has [M–H]⁻ = $m/z$ 185 and a neutral formula of $C_9H_{14}O_4$, with a diacid functionality proposed. However gas-phase formation of N11 (Supplement Sect. S2) is only possible through a hydroperoxide pathway that has been shown by theoretical calculations to be uncompetitive (Mackenzie-Rae et al., 2016). Furthermore, whilst compound N11 shows consistency with the tandem spectrum, a loss of 88 Da that may be expected for a

diacid is not observed. Therefore the proposed structure of compound N11 is only tentative, and potentially involves un-considered surface or condensed phase chemistry.

Compound N13 is present in all chromatograms as a major peak with a retention time of 9.92 (except for experiment 11 which had NO₂ added) and has [M–H]⁻ = $m/z$ 183 and a neutral formula of $C_{10}H_{16}O_3$. The tandem mass spectrum is relatively

uninformative, being dominated by a fragment ion at $m/z$ 139, suggesting that loss of the carboxylic acid group results in a stable ion. Given this information, compound N13 is ascribed to the first-generation acids (Table 2), with the exact isomer unable to be determined. First-generation acids are predicted to be major first-generation products in the gas-phase (Mackenzie-Rae et al., 2016), and so their detection in the particle phase is not surprising given their low vapour pressures.

Compound N12, observed in all chromatograms, has a [M–H]⁻ = $m/z$ 199 and is assigned a neutral formula of $C_{10}H_{16}O_4$. As discussed in section 3.5, the peak area of this species is reduced in the presence of a SCI scavenger. Only two main fragment ions are formed (loss of $CO_2$ and $CO_2+H_2O$) and further fragmentation using $MS^3$ yielded no additional ions. Two potential structures have been proposed. The first is a compound formed through the hydroperoxide channel undergoing a 1,6-H shift isomerisation, as proposed in Mackenzie-Rae et al. (2016), with a schematic shown in Fig. 4. However this pathway would

not be inhibited by the addition of a CI scavenger, as it is predominantly formed through re-arrangement of excited CIs. Alternatively, N12 can be attributed to a diacid. One possibility is that the diacid forms through abstraction of the aldehydic hydrogen by the CI, as shown in Fig. 4. The mechanism however, which was first proposed by Ma et al. (2007) to explain pinonic acid formation from a-pinene, was shown to be uncompetitive with competing pathways by computational calculations (Mackenzie-Rae et al., 2016). An alternative prospect is that aldehyde groups are oxidised to carboxylic acids,

increasing the mass by 16 Da, in a process that has been proposed to occur non-negligibly in the aerosol phase (Walser et al., 2008). For example, first-generation acids such as N13 could be oxidised to diacids. Therefore the current proposal of compound N12 remains tentative. Similarly, compound N9, observed at a retention time of 7.51 minutes, can form through oxidation of the carbonyl group in N12 to an acid. N9 is observed as a minor product in the majority of chromatograms, with a [M–H]⁻ = $m/z$ 215 and a neutral formula of $C_{10}H_{16}O_5$. The tandem mass spectrum is supportive of a diacid. Alternatively,



the same product can be formed following the same pathway as N12 with an additional 1,7-H shift between the acyl groups before radical termination, as shown in Fig 4.

### 3.1.2 Positive mode

Chromatograms produced from positive mode analysis are reasonably consistent across the experimental dataset showing
similar major product peaks, with the chromatogram from a representative experiment shown in Fig. 5. Tandem mass spectra from the positive mode however offers little structural insight, with identity of the major product peaks listed in Table 3 tentatively assigned based on mass to charge ratios and mechanistic insight (Mackenzie-Rae et al., 2016, 2017a). Preference is given to structures containing at least one carbonyl group, consistent with positive mode ESI charging mechanics. Where overlap is observed with negative mode spectra, the same species are reassigned. For example, compound N8 is found in the
positive mode at a similar retention time of 7.17 minutes, with $[M+Na]^+ = 183$. Given this, the P3 hydroperoxide containing isomer shown in Table 2 is assigned to the peak at 8.75 minutes. A corollary is that acidic isomers detected in the negative mode are unlikely to be responsible for peaks in the positive mode if corresponding retention times are significantly different.

A dominant feature of all positive mode chromatograms is the large peak at retention time 5.06 minutes, which has $[M+Na]^+ = m/z$ 151, and is assigned a neutral formula of $C_7H_{12}O_2$. This species is assigned to the major second-generation product P1, a dicarbonyl species. Similarly, compound P2 ($[M-H]^- = m/z$ 137, $C_6H_{10}O_2$) is also a major dicarbonyl second-generation product, with P1 and P2 having both been detected in the gas-phase (Mackenzie-Rae et al., 2017a). Estimated vapour pressures support their primary residence in the gaseous phase (Fig. 6). However, given their expected high gas-phase
concentrations, coupled with high aerosol loadings for most experiments inside the reactor, it is not unreasonable to assume enhanced partitioning of these prominent intermediate-volatile gas-phase species (Leungsakul et al., 2005; Walser et al., 2008). In addition, they may be present as a result of decomposition of larger species during the analysis.

The heaviest two products observed in the positive ionisation mode, P7 and P8, are attributed to peroxide containing
products from the hydroperoxide channel that have undergone additional intramolecular hydrogen abstractions and molecular oxygen additions. Termed autoxidation, this process has been used to explain aerosol formation from the reaction of monoterpenes with ozone in relatively clean environments, as it can result in the formation of compounds of extremely low volatility (Ehn et al., 2014; Jokinen et al., 2015). As shown in Fig. 6, the saturation vapour concentrations of P7 and P8 are not sufficient for nucleation, however their presence in the filter samples suggests autoxidation is occurring during α-
phellandrene oxidation, with it entirely possible that, upon further intramolecular H-transfers, species of the volatilities discussed in Ehn et al. (2014) and Jokinen et al. (2015) may be formed.



### 3.2 Product vapour pressures

Whilst the SOA analysed consists of a complex mixture of compounds, chromatographic analysis suggests that the SOA is dominated by a small number of major constituents. To assess the validity of identified products of differing functionalities as major aerosol constituents, saturation vapour concentrations ($C^*$, μg m$^{-3}$) for all predicted compounds were estimated

(method is described in Supplement Sect. S3) with results given in Tables 2 and 3, and plotted in two-dimensional volatility oxidation space in Fig. 6. Vapour pressures were found to span almost 10 orders of magnitude, indicating considerable variability in volatilities of proposed particle-phase products. At the organic particle mass loadings found during the chamber experiments (20 – 660 μg m$^{-3}$), organic compounds of intermediate volatility (IVOCs) are expected to primarily reside/partition into the gaseous phase, and therefore be minor contributors to the particle phase (Donahue et al., 2012).

Compounds classified as semi-volatile organics (SVOCs) are likely to have sizeable mass fractions in both the gaseous and aerosol phase at mass loadings encountered during chamber experiments, whilst those low volatility organic compounds (LVOCs) are thought to exist predominantly in the particle phase (Donahue et al., 2006, 2012).

Figure 6 shows the most prominent peaks detected in both the positive and negative mode chromatograms (namely *m/z* 160,

176, 184 and 200) are IVOCs and SVOCs. Three products were determined to be of low volatility, with these all $C_{10}$ species with between 5 and 7 oxygen atoms, whilst both first- and second-generation species are classified as SVOCs. Of the 21 products or product groups identified, 11 are $C_7$ or smaller second-generation products, with the increase in volatility associated with losing at least three carbons compensated in part by the decrease in volatility resulting from increased functionalisation, as shown in Figure 6. For both first- and second-generation products there is a clear negative correlation

between $C^*$ and O/C, with an offset due to the size of the hydrocarbon backbone between generations.

Figure 6 shows that none of the major particle phase species detected are classified as having extremely low volatility (ELVOCs), which has been argued as being a necessary requirement for compounds to homogenously nucleate upon supersaturation (Donahue et al., 2013; Heaton et al., 2007). The detected compounds are therefore unlikely to be responsible

for the rapid burst of freshly nucleated aerosol observed upon α-phellandrene reacting with ozone (Mackenzie-Rae et al., 2017a), but rather they progressively condense onto the nucleated core once aerosol clusters have grown past a critical size. For the ozonolysis of other monoterpenes, ELVOC formation has been proposed to occur through gas-phase accretion reactions (Bateman et al., 2009; Camredon et al., 2010; Heaton et al., 2007, 2009; Lee and Kamens, 2005; Tolocka et al., 2004) and autoxidation processes (Ehn et al., 2014; Jokinen et al., 2015). Higher mass compounds possibly formed as a

result of these reaction pathways were detected in both positive and negative mode spectra, and are discussed further in the following section.



### 3.3 High-resolution mass spectral analysis

In order to investigate the overall composition of the SOA, an average mass spectrum was produced of the section of the chromatogram where SOA components eluted (*i.e.* not including the dead volume or the final gradient equilibrium period). Figure 7a shows a representative mass spectrum in the *m/z* range 50 – 400 obtained in the positive mode (experiment 10).

The spectrum contains over 280 peaks with intensities exceeding 0.5% of the most abundant peak at *m/z* 223. An example negative mode spectra is shown in Fig. 7b, and contains over 200 peaks in the *m/z* range 50 – 450 with intensities exceeding 0.5% of the most abundant peak at *m/z* 199. In both spectra, peaks are predominantly clustered into wide groups separated by 14 amu ($CH_2$). The two spectra are explicitly compared in Fig. 7c by subtracting $^{23}$Na from the positive mode spectrum and adding $^1$H to the negative mode spectrum, making the plot pertinent to the neutral analytes. Depending on the experiment,

either the peak at *m/z* 160, 176 or 200 is the most intense. The compounds at MW = 200 Da correspond to the dominant first generation SOA products at $C_{10}H_{16}O_4$ (N12, P6 isomers), whereas the compounds at MW = 160 and 176 Da correspond to the dominant second generation SOA products at $C_7H_{12}O_4$ (N8, P3 isomers) and $C_7H_{12}O_5$ (N4 and N5) respectively. Therefore the ratio of these is a reflection of the degree of oxidation that had occurred when the filter samples were collected. Other abundant peaks routinely found in both the positive and negative ion mode spectra include *m/z* 144, 186 and

216, corresponding to products N6/N7, N11 and N9 respectively. There are some differences in the two spectra as a result of different ionisation efficiencies of SOA components. In addition, the larger downward peaks in the negative mode at *m/z* 156 and *m/z* 140 in Fig. 7c, are actually the result of a loss of $CO_2$ due to *in-source* fragmentation from SOA components at MW = 184 Da and MW = 200 Da. This highlights that caution is needed if a direct ESI with no prior separation approach is used. The assigned chemical formulas were also used to generate Kendrick plots, shown in the Supplement (Sect. S4), to

investigate homologous families of molecules based on either $CH_2$ ($KM_{CH2}$) or O ($KM_O$) units. Although the spectral distribution is similar in both modes, there are sufficient differences that analysis of both ionisation modes is necessary for complete characterisation of SOA composition.

### 3.4 Analysis of dimer products

A recognisable feature of the mass spectra in both modes in Figure 7 is the presence of monomers up to about *m/z* 250,

followed by dimers from *m/z* 250 – 500, corresponding to two oxygenated α-phellandrene product units. Similar clustering of peaks in mass spectra from SOA generated by monoterpene ozonolysis has been extensively reported (Bateman et al., 2009; Camredon et al., 2010; Heaton et al., 2007; Reinhardt et al., 2007; Tolocka et al., 2004; Walser et al., 2008). The intensity of the oligomeric signals are lower than the major monomeric peaks, and remain fairly consistent across the dimer domain. Proposed formulas range from $C_{12-20}H_{18-34}O_{5-11}$. Whilst relative intensities are lower than monomeric species, dimers

by nature contribute more mass and so are likely to have an important impact on the aerosol phase, especially in nucleation and early growth processes. Metrics such as O/C and H/C ratios, Kendrick mass defects and the double bond equivalency index (DBE) can be used to identify structural similarities and functionalities of SOA samples.





Species identified in the positive and negative modes, using the average mass spectra in Figure 7, are plotted in Van Krevelen space in Fig. 8. The majority of products have O/C ratios between 0.3 and 0.7, and H/C ratios between 1.5 and 1.7. The species detected show significant variation in elemental composition, although no significant differences in the distribution between products detected in positive and negative modes was observed. Furthermore, the distribution of monomer and oligomer species is similar, with oligomers having a similar average O/C but a narrower O/C range.

Using assigned chemical formulas, DBEs were calculated for all major spectral peaks (RI > 5%). Peaks containing an odd mass were taken as corresponding to molecules with one $^{13}C$ atom and were subsequently excluded. Calculated DBE values are plotted against carbon number in Fig. 9, with markers scaled by peak intensity (from SOA collected in experiment 10). The DBE values for major spectral components in the positive and negative modes were found to range from 2 – 5, with the largest peaks being monomers with a DBE of 2 or 3. The results suggest that most first-generation $C_{10}$ products likely contain one C=C double bond and two C=O groups, whilst saturated second-generation $C_7$ species contain two to three C=O bonds.

Heavier oligomeric species have a DBE ranging from 3 to 5. A clear split can be seen in the SOA composition between monomers containing 10 or less carbons and oligomers. The smallest dimers at $C_{12}$ are likely the result of reaction between two $C_6$ monomers, whilst the group at $C_{13}$ is likely the result of reaction between either a $C_6$ and $C_7$ monomer, or a $C_{10}$ and a $C_3$ product, formed from reaction at the second double bond. A further, more numerous group at $C_{15}$–$C_{20}$ is also seen and is likely a combination of a wider variety of $C_7$-$C_{10}$ species. Considering the DBEs of the base monomeric species, dimeric DBEs are consistent with oligomer formation through SCIs, peroxyhemiacetal and hemiacetal mechanisms. This is because the cumulative DBE of two monomers decreases by one upon accretion through these mechanisms. In contrast, for oligomerisation pathways involving dehydration, such as aldol condensation and esterification, the DBE is additive when going from momomers to a dimer. Given the prominence of monomeric species with a DBE of 3, one would expect to see dimeric products with a DBE of 6 if condensation pathways were significant. Of the remaining considered accretion pathways, hemiacetal formation has previously been found to be thermodynamically unfavourable (Barsanti and Pankow, 2004), whilst peroxyhemiacetals are likely to be thermally labile and thus potentially unstable when subject to high temperatures during ESI ionisation (Camredon et al., 2010; Surratt et al., 2006). It is therefore concluded that oligomerisation through bimolecular reaction of SCIs is likely to be the major accretion process from α-phellandrene ozonolysis under chamber simulation conditions.

Table 4 lists the mass of potential dimers that could be formed from the reaction of select SCIs with first- and second-generation products. The majority of the proposed masses from these bimolecular addition reactions were detected in the positive and negative ionisation mode mass spectra. In the literature, Bateman et al. (2009) proposed that the dominant



oligomerisation mechanism in the ozonolysis of limonene is the reaction between SCIs and stable first-generation products, whilst Lee and Kamens (2005), Hall and Johnston (2012) and Kristensen et al. (2016) showed that reactions of SCIs with first-generation carboxylic acids are sources of dimers in the ozonolysis of α-pinene. For isoprene ozonolysis, Sakamoto et al. (2017) argued that oligomerisation occurs through an initial reaction of an SCI with an organic acid, followed by

sequential insertion of SCIs. This mechanism has support from their prior study into the ozonolysis of ethylene (Sakamoto et al., 2013). Meanwhile, for linear alkenes and enol ethers, Sadezky et al. (2006, 2008) and Zhao et al. (2015b) proposed that oligoperoxides, formed from the addition of SCIs to organic peroxy radicals, are responsible for SOA formation and growth.

In addition to formation from SCIs, numerous other accretion processes have been proposed in the literature to account for

particle-phase oligomers including peroxyhemiacetal formation, hemiacetal formation, aldol condensation and acid anhydride and ester formation, which are summarised in reviews by Hallquist et al. (2009), Kroll and Seinfeld (2008) and Ziemann and Atkinson (2012). These processes have been proposed to occur in the gas-phase, heterogeneously through reactive uptake on the surface of aerosols and/or *via* two condensed monomers. SOA products were detected at the *m/z* values of dimers predicted to be formed *via* these pathways, with different monomer combinations, and accretion pathways,

resulting in multiple dimeric products of the same mass. So whilst it is likely that the reaction of SCIs is important in forming dimers, significant contributions from other accretion channels cannot be readily discounted, with further investigation necessary to determine the relative importance of discussed dimerisation/oligomerisation channels.

### 3.5 Impact of SCI scavenging on SOA composition

Over the last decade, many studies have shown that SCI are important precursors to SOA formation, using a range of SCI

scavengers to investigate effects on yields, masses, number and composition of the particles formed. Recent results from Sakamoto et al. (2017) showed that SOA formation from isoprene ozonolysis was suppressed at high RH due to scavenging of the $CH_2OO$ SCI through rapid reaction with water. However, a portion of the SCIs formed had lower reactivity towards water, mostly likely one or more of the corresponding $C_4$ SCI formed upon ozonolysis (Newland et al., 2015). In contrast, Kristensen et al. (2014) found that new particle formation and the amount of dimer-type molecules formed during α-pinene

ozonolysis increased at higher RH values. Bonn et al. (2002) reported a significant decrease in new particle formation and aerosol yields at higher RH values for the ozonolysis of exocyclic monoterpenes, with number concentration only moderately affected and no significant change in yield observed for the ozonolysis of endocyclic alkenes. Recent work from Newland et al. (2017) has shown that the reaction of certain $C_{10}/C_9$ monoterpene SCIs with water are slower than that for $CH_2OO$. Therefore, the reactivity of SCI with water is structurally dependent and may determine the effect of relative

humidity on reducing SOA formed *via* SCI oligomerisation.

Zhao et al. (2016) found that particle number and higher molecular weight products formed from the ozonolysis of α-cedrene, a $C_{15}$ sesquiterpene, were significantly reduced upon addition of formic acid, HC(O)OH, as a SCI scavenger.



Ahmad et al., (2017) have recently shown that adding acetic acid or acetone to limonene ozonolysis experiments significantly delayed the formation of particles leading to reduced SOA masses and a reduction in particle numbers, with acetic acid showing the strongest effect. These findings are consistent with the study of Bonn et al. (2002), who showed the addition of formic acid to have a significant inhibiting effect on nucleation and on total aerosol volume for the ozonolysis of

5 β-pinene, with similar but less drastic changes reported for α-pinene. Similar observations were also reported in the companion paper using formic acid as an SCI scavenger in the α-phellandrene ozonolysis experiments, with SCI scavenged experiments showing a large reduction in initial particle number concentrations, suggesting a reduction in the number of SOA-nucleating agents, and lower total SOA yields (Part 1, Mackenzie-Rae et al., 2017a).

In order to investigate the impact of the SCI on the formation of dimers and SOA composition more generally, a series of experiments were carried out with high concentrations of formic acid (Table 1, experiments 6 and 7). Figure 1 shows the main differences in the chromatograms from SCI scavenged experiments were an increase in the relative amount of compounds N10 and N7, and a decrease in the amount of compounds N12 and N13. Compound N10 has $[M–H]^- = m/z$ 173 and is assigned a neutral formula of $C_8H_{14}O_4$. It is a relatively minor species in the majority of filter samples, however is

observed as a major peak at retention time 7.96 minutes in the chromatograms of SOA produced with formic acid added as a SCI radical scavenger (experiments 6 and 7). Tandem mass spectra offer some insight, with losses of 44 Da and 62 Da (carboxylic acid and another oxygenated functional group). A small fragment ion at $m/z$ 83 ($C_5H_7O$) indicates a carbonyl or acid functionality in the α-position to the branched side chain. The structure contains 8 carbons, suggesting it is formed *via* reaction of a $C_7$ second-generation product and formic acid. The scavenging mechanism of SCI by formic acid is known to

form a hydroperoxy methyl formate (HPMF) for the $CH_2OO$ SCI (Neeb et al., 1995; Hasson et al., 2001; Sakamoto et al., 2013). Horie et al. (1997) have detected l-hydroperoxyethyl formate (HPEF) using FTIR spectrometry in *cis/trans*-but-2-ene ozonolysis experiments in the presence of formic acid, which decomposes to acetic formic acid anhydride. Liu et al. (2015) have also looked at the reaction of a range of alkyl substituted SCI with deuterated formic acid and acetic acid using photoelectron spectroscopy, directly detecting a range of partially deuterated alkyl substituted vinylhydroperoxide species

formed through an acid-catalysed tautomerisation reaction. Complementary theoretical calculations predict several reaction pathways, including a barrierless insertion reaction to yield a hydroperoxyalkyl formate (Liu et al., 2015, Kumar et al., 2014). An equivalent functionalised hydroperoxyalkyl formate is shown in Fig. 10 for a major second generation SCI (Mackenzie-Rae et al., 2017a). However, the established mechanism cannot explain the formation of the observed product, as dehydration to an acid anhydride species (Neeb et al., 1995; Horie et al., 1997) forms a product with a mass 2 Da too low.

An alternative decomposition process is therefore required to explain formation of N10 from the major second-generation CIs. It is therefore proposed that formic acid can additionally participate in accretion reactions with products, driven by the high concentrations of formic acid inside the reactor. An example is given in Fig. 10, yielding the predicted product through a hemiacetal-type reaction (c.f. Hallquist et al., 2009). Whilst the proposed product could potentially match the MS/MS spectrum, the thermodynamics of its formation remain uncertain. Because aerosol yields were observed to decrease upon the




addition of formic acid (Mackenzie-Rae et al., 2017a), the net effect of the increased product contribution of N10 to the condensed phase is less than that which originates from SCIs in non-scavenged experiments.

There is also a more pronounced peak at R.T. 6.23 minutes (N7) in the formic acid experiments. The two peaks at R.T = 6.01 min and R.T. = 6.23 min are structural isomers of $C_7H_{12}O_3$ with the acid and carbonyl functionalities on different sides of the molecule. The $MS^2$ spectra are almost identical (loss of 44 and 46 Da, see Supplement Figs. S1.6 and S1.7), with the peak at 6.23 having a small additional peak from loss of CO (28 Da), suggesting the carbonyl is in the α-position to the branched chain. The structural similarity and enhanced formation of both N7 and N10 upon addition of formic acid suggests that there may be an intrinsic relationship between the two species. One possibility is that N7 is a decomposition product of N10 or

other dimeric species formed through SCI scavenging.

The two major peaks that have significantly reduced peak intensities are both $C_{10}$ species, containing carbonyl and acidic functionalities (N12 = $C_{10}H_{16}O_4$ and N13 = $C_{10}H_{16}O_3$). This result further suggests that these structurally similar species are related, with N12 being formed from further oxidation of N13. The absence of a N13 peak when $NO_2$ is present suggests that

formation is inhibited by $NO_2$, which would also scavenge SCIs to form carbonyls. Interestingly, N13 is only moderately impacted when using formic acid as the SCI scavenger, implying that scavenging by formic acid may ultimately result in evolution of acid functionality, either through further degradation or catalysing CI tautomerisation (Kumar et al., 2014; Liu et al., 2015).

The experiments with formic acid were both carried out at low initial a-phellandrene mixing ratios (<20 ppb). The average mass spectra for scavenged and non-scavenged experiments carried out at very low concentrations do not show the obvious dimer region seen in Figure 7, most likely as a result of increased background interferences. In order to assess the change in the oligomer composition, the extracted ion chromatograms were compared for two experiments with and without formic acid but with similar levels of initial VOC (experiments 3 and 7). The change in the relative *m/z* peak areas associated with

the products of reactions of the main SCIs with a range of stable carbonyl and acids for scavenged and unscavenged experiments were estimated and are shown in Table 4. A direct comparison of peak areas is complicated due to a) a large difference in the amount of SOA mass produced (121 versus 28 mg m$^{-3}$), which will change the volatility distribution of SOA, b) differences in ozone levels in the chamber, and c) the fact that dimers from different reaction pathways can result in dimers of the same mass. However, there are clear differences in the distribution of dimers between the two experiments.

Seven dimer mass peaks [MW = 288, 304 ($C_{14}$), 312, 328, 344, 352, 368 Da, bold numbers in Table 4] showed a reduced peak area when the scavenger was used. Four dimer masses [MW = 256, 270, 302, 304 ($C_{13}$) Da, italicised in Table 4] were more abundant when the scavenger was used. The remainder of masses in Table 4 show either similar peak areas or have inconsistencies, for example the extracted ion chromatogram of m/z 341 [M = 342 Da] has two peaks, one that is similar in both samples and one that is reduced in the presence of formic acid. The number of high mass oligomers was also





significantly suppressed upon addition of $NO_2$ to the system, with $NO_2$ also acting as a sink for SCIs (Welz et al., 2012).

The observed change in oligomeric species distributions upon the addition of a SCI scavenger may help to explain macroscopic observations, where the number of aerosol particles observed during the initial stages of experiments were found to decrease upon addition of formic acid (Mackenzie-Rae et al., 2017a). The measured reaction rates of $C_1$ and $C_2$ SCI with formic acid have been measured to be very fast, in excess of $1 \times 10^{-10}$ cm$^3$ s$^{-1}$, several orders of magnitude faster than reaction with water (Welz et al., 2014). Given that the concentrations of water inside the reactor was three orders of magnitude larger than the concentration of formic acid (Table 1), the results presented here for α-phellandrene suggest that scavenging of the SCI by small carboxylic acids is much faster than the sink reaction with water, and can lead to a significant drop in the amount of dimers and hence SOA that is formed, with potentially important atmospheric implications (Sipilä et al., 2014). Further work is needed to gain a holistic understanding of SCI chemistry, principally in identifying the chemical pathways that lead to dimer formation from SCIs and characterising the chemical impact of SCI scavengers such as water, $NO_x$, $SO_2$ and oxygenated volatile organic compounds (i.e. acids, carbonyls, alcohols) on atmospheric aerosol composition.

## 4 Conclusion

The compositional components of the organic aerosol formed from the gas-phase ozonolysis of α-phellandrene were identified and characterised for the first time, using a high-resolution Quadrupole-Orbitrap mass spectrometer. In total 21 products or product groups were identified from chromatograms, aided by tandem mass spectrometry, with electrospray ionisation used in both positive and negative modes to gain complementary compositional information. Both polyfunctional first- and second-generation products were found to be prominent in the aerosol, with products significantly more complex than those proposed from basic gas-phase chemistry in the companion paper (Mackenzie-Rae et al., 2017a). The results therefore suggest that a number of first-generation unsaturated products are sufficiently volatile to remain in the gas-phase and further react with ozone, with many of the resultant second-generation products able to partition into the particle-phase despite saturation vapour concentration estimates classifying a large number of products as semi-volatile. Oligomeric species are observed in both positive and negative mode spectra, with evidence supporting dimerisation through bimolecular reactions of stabilised Criegee intermediates. Suppression of oligomer spectral peaks upon the addition of a stabilised Criegee intermediate scavenger coincided with reduced experimental SOA growth and formation, with stabilised Criegee intermediate dimers, and/or higher order oligomers, believed to be a significant source of the highly condensable products required for the rapid nucleation and growth of new particles observed in the system (Mackenzie-Rae et al., 2017a). Further investigation is required however to better characterise the role of stabilised Criegee intermediates in the system, and to parameterise the specific accretion mechanisms occurring. Nonetheless this study has provided the first understanding of the nature of SOA formed from the ozonolysis of α-phellandrene, and more generally contributes to an enhanced knowledge of




the atmospheric effects and implications of monoterpene emissions.

## Data availability

Labelled tandem mass spectra for all identified products are provided in the Supplementary Information. Data files containing positive and negative stick spectra with prescribed OC, HC, DBE and KMD values have been archived, with the

files publically available at: www.pure.york.ac.uk (doi will be assigned when paper is accepted.).

## Acknowledgements

The authors would like to thank Tengyu Liu, Wei Deng, Zheng Fang and Yanli Zhang for their assistance in collecting the filter samples. FMR acknowledges support from the Wolfson Atmospheric Chemistry Laboratories in the Department of Chemistry at the University of York for a 2 month visiting position in order to collaborate on the filter sample data analysis

and the UWA Postgraduate Student Association and Convocation of UWA Graduates for travel funding. Chamber experiments were made possible through funding by the Strategic Priority Research Program of the Chinese Academy of Sciences (Grant No. XDB05010200); Ministry of Science and Technology of China (Grant No. 2016YFC0202204); and National Natural 10 Science Foundation of China (Grant No. 41530641/41571130031). The Orbitrap-MS was funded through a UK Natural Environment Research Council Capital Grant (NERC Grant CC090).

*Competing interests.* The authors declare that they have no conflict of interest

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



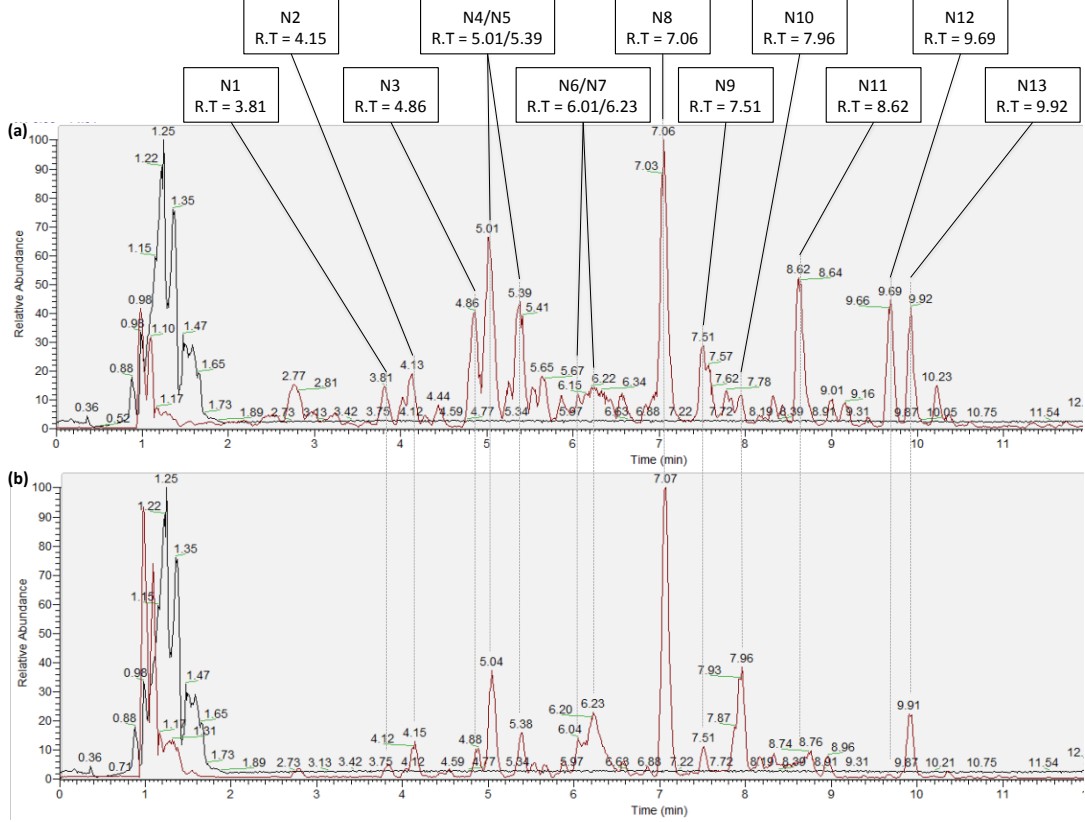

**Figure 1: Negative ionisation mode chromatogram of SOA generated from the ozonolysis of α-phellandrene in (a) a standard experiment and (b) an experiment that had formic acid added as a stabilised CI scavenger. Red line represents sample whilst black line is a preconditioned blank filter. Major negative mode ionisation peaks detected across all experiments (Table 2) are labelled.**





**Figure 2: Simplified ozonolysis pathway outlining the formation of some major species detected in negative mode ESI. Analogous structures can additionally be traced by varying the order of addition to the two double bonds and/or which CIs participate in the hydroperoxide (HP) and stabilisation channels respectively.**

**Figure 3: Proposed formation mechanism of compound N4. N5 is formed through an analogous mechanism starting with initial attack of ozone at the more substituted double bond.**




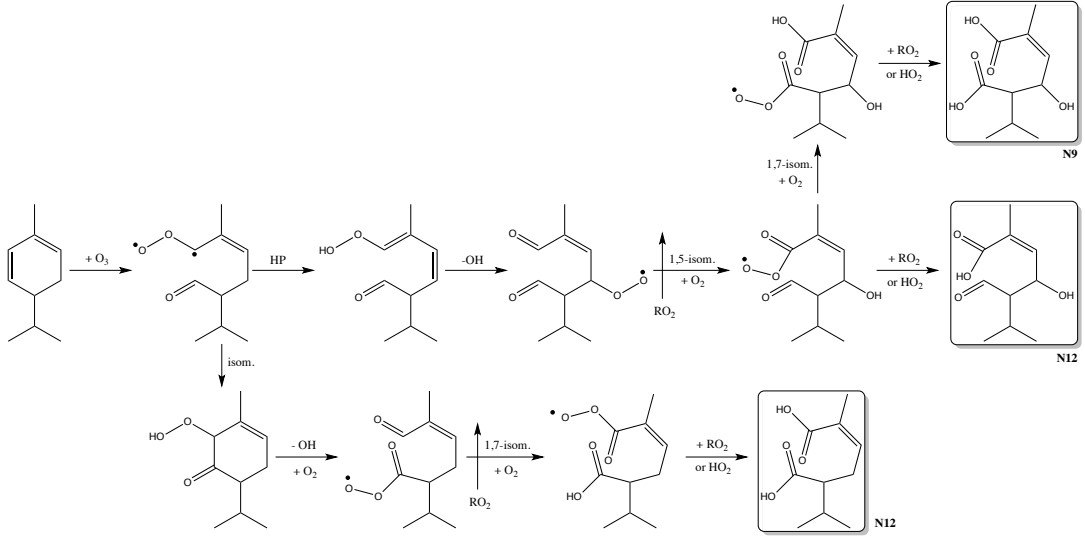

**Figure 4: Formation mechanisms of compounds N9 and N12.**

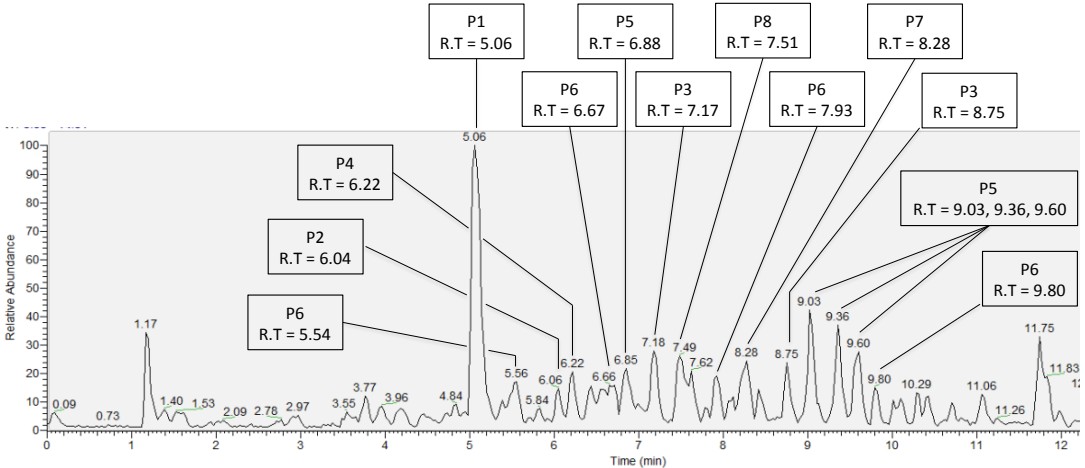

5  **Figure 5: Positive ionisation mode chromatogram of SOA generated from the ozonolysis of α-phellandrene. Major positive mode ionisation peaks detected across all experiments (Table 3) are labelled.**





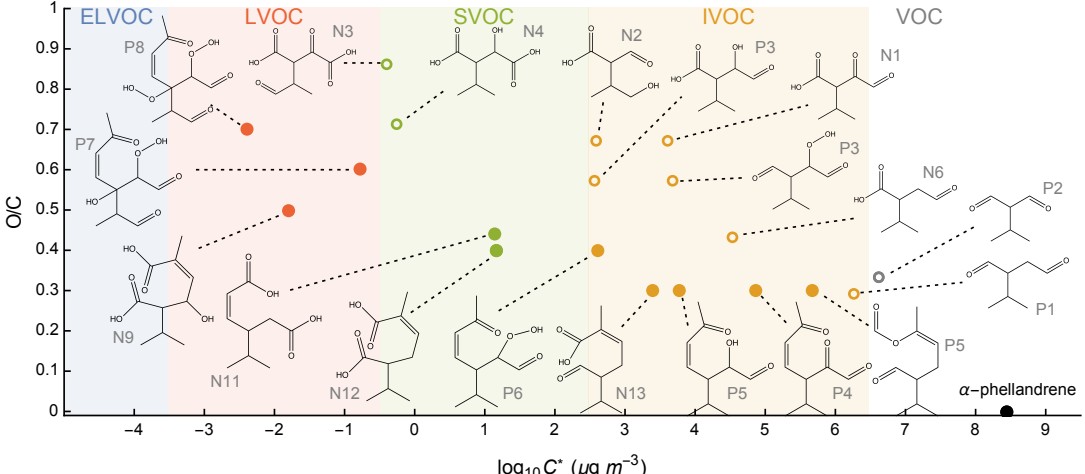

**Figure 6: α-phellandrene and detected particle phase ozonolysis products from both positive and negative ESI modes plotted in two dimensional volatility-oxidation space, constrained by saturation concentration on the x-axis and O/C elemental ratios on the y-axis. Molecular structures of products are shown. Filled and open circles denote first- and second-generation products respectively, with coloured regions indicating broad volatility classes. Volatility distribution can be compared with gas-phase species through Fig. 9 in Mackenzie-Rae et al. (2017a).**





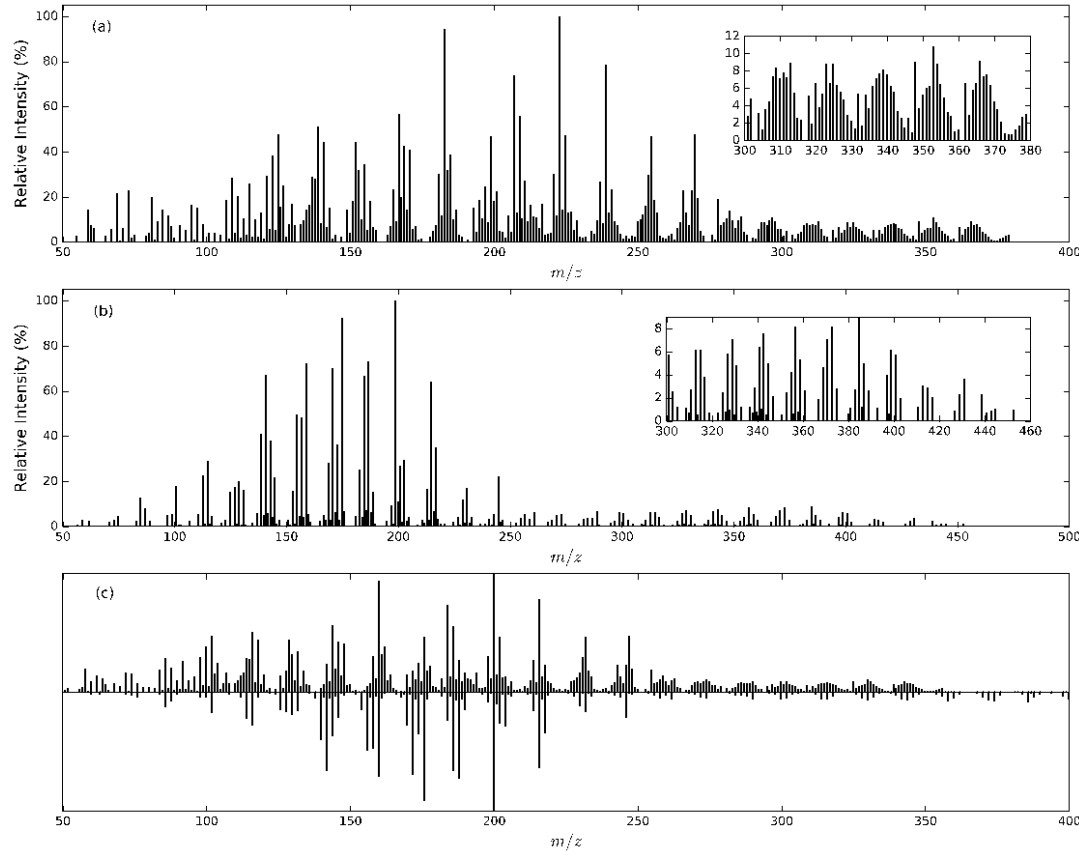

**Figure 7: Representative ESI mass spectra from experiment 10 showing peaks with ≥ 0.5% abundance relative to the largest peak in the spectrum for stick spectra in the (a) positive and (b) negative modes. Comparison plot (c) shows mass spectra after subtracting $^{23}$Na from the positive *m/z* scale (up) and adding $^{1}$H to the negative *m/z* scale (down).**





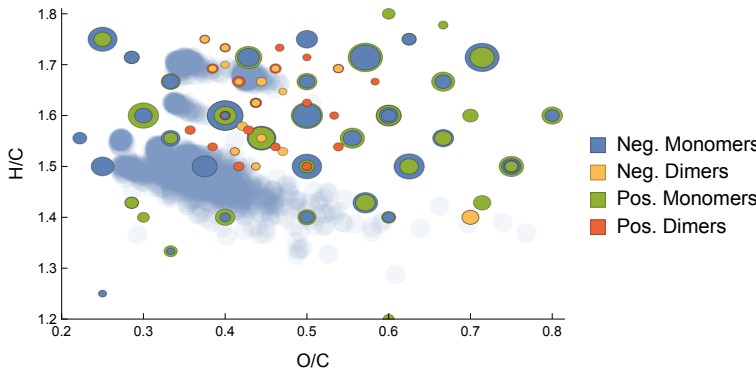

**Figure 8: Van Krevelen plot comparing online experimental measurements by an AMS (blue shading) (Fig. 18 in Mackenzie-Rae et al., 2017a) with assigned formulas from the ESI mass spectra (RI > 5%) in the negative mode (blue ≤ C$_{10}$, yellow > C$_{10}$) and positive mode (green ≤ C$_{10}$, red > C$_{10}$). Circle size is proportional to relative signal intensity in either mode.**

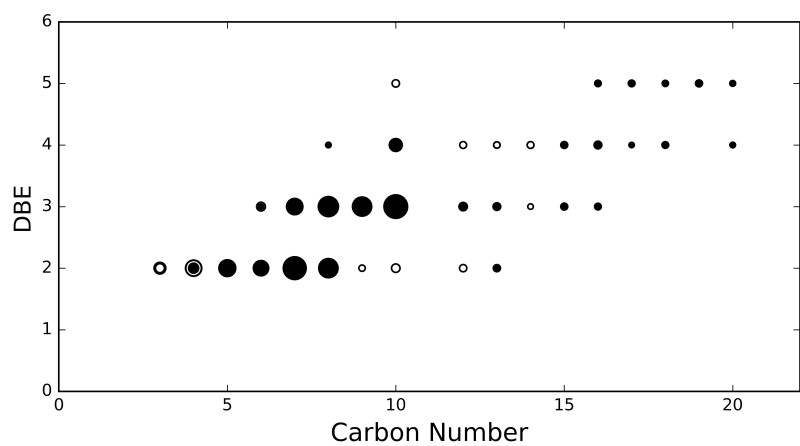

**Figure 9: DBE values plotted against carbon number for ions (RI > 5%) detected in the positive (open circle) and negative (closed circle) ionisation mode spectra from experiment 10. The size of data points is proportional to signal intensity.**







**Figure 10: Reaction of (a) prominent second-generation CI with formic acid and (b) accretion of major second-generation carbonyl with formic acid to yield proposed structure for compound N10. An unknown channel is required to join pathway (a) with product N10.**

| Exp No.[a] | Temp. (K) | RH (%) | VOC[b] (ppb) | O$_3$ (ppb) | Additives | Total SOA Mass ($\mu$g m$^{-3}$)[c] | SOA Mass Collected ($\pm$0.01, mg) |
|---|---|---|---|---|---|---|---|
| 1 | 297.1 $\pm$ 0.4 | 2.5 $\pm$ 0.6 | 19 $\pm$ 7 | > 259 | Cyclohexane | 86 $\pm$ 9 | 0.16 |
| 2 | 297.5 $\pm$ 0.5 | 2.1 $\pm$ 0.7 | 10 $\pm$ 4 | > 86 | Cyclohexane | 21 $\pm$ 2 | 0.05 |
| 3 | 297.2 $\pm$ 0.2 | 2.3 $\pm$ 0.6 | 21 $\pm$ 8 | > 83 | Cyclohexane | 121 $\pm$ 13 | 0.21 |
| 4 | 297.4 $\pm$ 0.5 | 2.2 $\pm$ 0.9 | 32 $\pm$ 13 | > 193 | Cyclohexane | 312 $\pm$ 33 | 0.25 |
| 5 | 297.6 $\pm$ 0.7 | 1.8 $\pm$ 0.4 | 29 $\pm$ 11 | > 114 | Cyclohexane | 178 $\pm$ 19 | 0.15 |
| 6 | 298.0 $\pm$ 0.3 | 1.6 $\pm$ 0.1 | 16 $\pm$ 6 | > 470 | Cyclohexane Formic acid | 52 $\pm$ 6 | 0.12 |
| 7 | 298.0 $\pm$ 0.1 | 1.9 $\pm$ 0.2 | 19 $\pm$ 8 | > 499 | Cyclohexane Formic acid | 28 $\pm$ 3 | 0.10 |
| 8 | 298.7 $\pm$ 0.6 | 5.2 $\pm$ 0.2 | 61 $\pm$24 | > 56 | Cyclohexane | 200 $\pm$ 21 | 0.50 |
| 9 | 298.5 $\pm$ 0.4 | 4.9 $\pm$ 0.4 | 67 $\pm$ 27 | > 101 | - | 341 $\pm$ 36 | 2.50 |
| 10 | 298.2 $\pm$ 0.5 | 4.8 $\pm$ 0.3 | 175 $\pm$ 69 | > 174 | Cyclohexane | 658 $\pm$ 70 | 0.70 |
| 11 | 298.1 $\pm$ 0.4 | 4.5 $\pm$ 0.2 | 88 $\pm$ 35 | > 132 | NO$_2$ (385 ppb) | 505 $\pm$ 53 | 0.74 |

[a]Refer to Mackenzie-Rae et al. (2017a).

[b] $\alpha$-phellandrene.

[c]Wall loss corrected.

**Table 1: Experimental conditions and SOA collection information for chamber experiments.**



| Product ID | Retention Time (min) | Measured $m/z$ | Neutral Formula | Possible Structures | $P_{vap}$ (atm) | $C^*$ ($\mu g\ m^{-3}$) |
|---|---|---|---|---|---|---|
| N1 | 3.81 | 157.05 | $C_7H_{10}O_4$ | | $6 \times 10^{-7}$ | $4 \times 10^{3}$ |
| N2 | 4.15 | 145.06 | $C_6H_{10}O_4$ | | $4 \times 10^{-8}$ | $4 \times 10^{2}$ |
| N3 | 4.86 | 187.06 | $C_7H_8O_6$ | | $4 \times 10^{-11}$ | $4 \times 10^{-1}$ |
| N4 | 5.01 | 175.06 | $C_7H_{12}O_5$ | | $6 \times 10^{-11}$ | $6 \times 10^{-1}$ |
| N5 | 5.39 | 175.06 | $C_7H_{12}O_5$ | | $1 \times 10^{-10}$ | $1 \times 10^{0}$ |
| N6 | 6.01 | 143.07 | $C_7H_{12}O_3$ | | $6 \times 10^{-6}$ | $4 \times 10^{4}$ |
| N7 | 6.23 | 143.07 | $C_7H_{12}O_3$ | | $6 \times 10^{-6}$ | $4 \times 10^{4}$ |
| N8 | 7.06 | 159.06 | $C_7H_{12}O_4$ | | $5 \times 10^{-8}$ | $5 \times 10^{2}$ |
| N9 | 7.51 | 215.09 | $C_{10}H_{16}O_5$ | | $3 \times 10^{-12}$ | $2 \times 10^{-2}$ |
| N10 | 7.96 | 173.08 | $C_8H_{14}O_4$ | | $8 \times 10^{-6}$ | $3 \times 10^{4}$ |
| N11 | 8.62 | 185.08 | $C_9H_{14}O_4$ | | $3 \times 10^{-9}$ | $1 \times 10^{1}$ |



| N12 | 9.69 | 199.08 | $C_{10}H_{16}O_4$ | | $2 \times 10^{-9}$ | $1 \times 10^1$ |
| N13 | 9.92 | 183.10 | $C_{10}H_{16}O_3$ | | $3 \times 10^{-7}$ | $2 \times 10^3$ |

**Table 2: Major species detected in negative mode ESI with postulated structures consistent with composition, the mechanism proposed by Mackenzie-Rae et al. (2017a) and tandem mass spectra. Estimated saturation vapour pressures ($P_{vap}$) and saturation vapour concentrations (C*) for each product are also listed. Proposed structure list is not exhaustive, with other isomers existing.**



| Product ID | Retention Time (min) | Measured $m/z$ | Neutral Formula | Possible Structures | $P_{vap}$ (atm) | $C^*$ ($\mu g\ m^{-3}$) |
|---|---|---|---|---|---|---|
| P1 | 5.06 | 151.10 | $C_7H_{12}O_2$ | | $2 \times 10^{-4}$ | $2 \times 10^6$ |
| P2 | 6.04 | 137.10 | $C_6H_{10}O_2$ | | $6 - 45 \times 10^{-4}$ | $4 - 35 \times 10^6$ |
| P3 | 7.17, 8.75 | 183.10 | $C_7H_{12}O_4$ | | $4 - 64 \times 10^{-8}$ | $3 - 48 \times 10^2$ |
| P4 | 6.22 | 205.08 | $C_{10}H_{14}O_3$ | | $4 - 12 \times 10^{-6}$ | $4 - 11 \times 10^4$ |
| P5 | 6.88, 9.03, 9.36, 9.60 | 207.10 | $C_{10}H_{16}O_3$ | | $0.3 - 700 \times 10^{-6}$ | $0.2 - 2930 \times 10^4$ |
| P6 | 5.54, 6.67, 7.93, 9.80 | 223.11 | $C_{10}H_{16}O_4$ | | $2 - 65 \times 10^{-9}$ | $1 - 40 \times 10^1$ |
| P7 | 8.28 | 253.10 | $C_{10}H_{14}O_6$ | | $2 \times 10^{-11}$ | $2 \times 10^{-1}$ |





| | | | |
|---|---|---|---|
| P8 | 7.51 | 269.11 | $C_{10}H_{14}O_7$ |

(structure) $5 \times 10^{-13}$  $4 \times 10^{-3}$

**Table 3: Major species detected in positive mode ESI with postulated structures consistent with composition and the mechanism proposed by Mackenzie-Rae et al. (2017a), along with estimated saturation vapour pressures ($P_{vap}$) and saturation vapour concentrations (C*). Proposed structure list is not exhaustive, with other isomers existing.**

| | | Stable Species | | | | | | | |
|---|---|---|---|---|---|---|---|---|---|
| | M.W | 72 | 128 | 144 | 160 | 168 | 182 | 184 | 200 |
| SCIs | 88 | ND | 216 | 232 | 248 | *256* | *270* | ND | **288** |
| | 104 | ND | 232 | ND | 264 | ND | 286 | **288** | *304* |
| | 144 | 216 | ND | **288** | **304** | **312** | 326 | **328** | **344** |
| | 158 | 230 | 286 | *302* | 318 | 326 | ND | 342 | ND |
| | 160 | 232 | **288** | 304 | 320 | **328** | 342 | **344** | 360 |
| | 184 | *256* | **312** | **328** | **344** | **352** | ND | **368** | 384 |

5  **Table 4: *m/z* of possible dimer products from α-phellandrene ozonolysis formed from the reaction of SCIs with non-radical ozonolysis products. Reaction of SCIs with carbonyls forms secondary ozonides whilst reaction with acids yields acyloxyalkyl hydroperoxides dimers. Masses in bold were reduced upon introduction of a SCI scavenger, masses in italics had increased abundance upon addition of a SCI scavenger and masses in normal font remained similar or were inconsistent upon introduction of a SCI scavenger. ND denotes that the proposed mass was not detected in the ESI mass spectra.**

