# Peer review of "Ozonolysis of $\alpha$ -phellandrene – Part 2: Compositional analysis of secondary organic aerosol highlights the role of stabilised Criegee intermediates"

_Atmospheric Chemistry and Physics, 2017_

## Referee Comment (RC1) · Anonymous Referee #2 · 19 Sep 2017

Review

The study by Mackenzie-Rae et al. presents interesting new results on the chemical composition of SOA from ozonolysis of alpha-phellandrene. In general the results are well presented and discussed, though it is somewhat disappointing that a manuscript authored by several native English-speaking persons contains a number of grammatical errors and unclear sentences. I recommend that the work is published after the following points have been adequately addressed.

[Figure]

Page 1 Abstract: The abstract must mention that the study only investigates water-extractable components of the SOA.

Page 1 Line 16: Why is quadrupole written with capital first letter? Since it is not used for an abbreviation, nor is it a name, lower case would be more correct.

Page 1 Line 29: "around Eucalypt forests regions" -> please correct the grammar here

Page 2 Line 13: "One monoterpene for which little study has been conducted is $\alpha$-phellandrene." -> please rewrite to make the sentence more clear and correct

Page 2 Line 17: " Believed then to be of particular importance is the abundance of $\alpha$-phellandrene in extracts of numerous species of Eucalypts" -> please rewrite to make the sentence more clear

Page 2 Line 28: "However, postulated gas-phase species could not explain the properties of the SOA observed." -> please make it clear that you refer to the companion paper here.

Page 3 Line 6: Please also refer to some of the first studies that applied this technique to aerosols.

Page 3 Line 8-9: I suggest to remove "and relevant to the current discussion"

Page 3 Line 25-30: The storage temperature is higher than often used in other studies (-18C). For how long were the samples stored and can you comment on the possible influence on the analytical results?

Page 3 Line 29: Why were the filters only extracted in water, not a polar solvent? It is important that this is emphasized throughout the manuscript - unless the authors have results showing that most of the SOA is actually extracted by this method. If so, those results or previous studies should be mentioned.

Page 4 Line 1: Was the effect of "evaporation to dryness" on molecular composition of samples investigated?

Page 4 line 6: "reverse-phase" should be corrected to "reversed-phase".

Page 4 line 9: Did you actually inject 30 mL? Please provide dimensions of the column and eluent flow rate.

Page 4 line 15: How frequently was the mass calibration done? Please be specific.

Page 4 Line 19-20: It seems strange that you mention methylation during extraction, when you extracted in water. The influence of adduct formation with methanol was further investigated by Bateman et al. (2008) and Kristensen and Glasius (2011), also showing that methanol esters can form during HPLC analysis.

Page 4 Line 23: "Because all first- and second-generation products of $\alpha$-phellandrene ozonolysis contain at least one functional group that is capable of ionisation (Mackenzie-Rae et al., 2016, 2017a), it is reasonable to assume that a high proportion of water soluble SOA components will be observed..". Especially the first part of the sentence is quite a bold statement, which should be modified. The next sentences on ionization must be shortened to avoid repetition.

Page 5 Line 19: "mass spectra are" or "mass spectrum is". Please correct.

Page 7 Line 9: Can you add a reference where the loss of 88 Da is observed from diacids?

Page 7 Line 18: Since the vapour pressures have not been measured, please change to "estimated low vapour pressures".

Page 7 Line 30: Has this process not been investigated since it was suggested in 2008?

Page 8 Line 21: "it is not unreasonable to assume enhanced partitioning of these prominent intermediate-volatile gas-phase species". What do you mean by enhanced? Compared to what?

Page 8 Line 30-31: The last sentence is quite unclear.

Page 9 Line 2-3: "Whilst the SOA analysed consists of a complex mixture of compounds, chromatographic analysis suggests that the SOA is dominated by a small number of major constituents." Please write more clearly what you mean - is it a complex mixture or a small number of major constituents?

Page 9 Line 5: Please mention the tool you used for this, since these estimates vary considerably.

Page 9 Line 15-16: Please clarify this sentence.

Page 10 Line 25-26: Please order these references by publication year.

Page 16 Line 6: Please add information on the reaction rate with water.

Additional reference: The study of Zhang et al. (2015) provided new information on molecular composition and dimers in monoterpene SOA and is suggested as an additional reference.

Figure 3: The first step needs additional information.

Table 1: It should be clear from the table caption, that most of this information is from the companion paper.

References: A.P. Bateman et al., Environmental Science and Technology (2008) 42, 7341-7346 K. Kristensen and M. Glasius, Atmospheric Environment (2011) 45, 4546-4556. Y. Zhang et al., Proc. Natl. Acad. Sci. (2015) 112, 14168-14173.

---

## Referee Comment (RC2) · Felix A. Mackenzie-Rae et al. · 15 Nov 2017

Ms.No. acp-2017-654

The authors describe experimental findings of aerosol analysis from the gas-phase reaction of ozone with α-phellandrene based on filter measurement. Analysis has been carried out by a HPLC-MS technique using an Orbitrap tandem mass spec. Reaction conditions were chosen in such a way that secondary chemistry influenced the product distribution. Reactant concentrations are in most cases clearly higher than atmospheric levels. Addition of formic acid, acting as sCI scavenger, repressed bimolecular and most likely unimolecular sCI pathways resulting in changing product distribution. According to that, the authors concluded that sCI reactions are important for the aerosol constituents detected.
From my perspective there are some points that should be considered before acceptance can be suggested.

1) The authors used filter measurements. Nothing is said regarding possible aerosol processing between sampling and analysis. What was the delay time between sampling and analysis?

2) What is the reason for the low RH < 5% in all runs?

3) Unfortunately, there is no pair of experiments with formic acid ON or OFF and otherwise constant reaction conditions. The best pair is Exp.1 and Exp.7 where merely ozone was changed by a factor of about 2. Please show for these experiments the corresponding spectra, especially in the "dimer" range, which should confirm author´s conclusion of the importance of bimolecular sCI reactions for SOA formation. Identified "dimers" as given in table 4 should be marked in the spectra. Please state in the figure caption of all given spectra the EXP. number according to table 1.

4) Experiments have been conducted with higher reactant concentrations compared with atmospheric conditions resulting in higher steady-state sCI (and other product) concentrations as those expected in the atmosphere. Consequently, bimolecular pathways are less important under atmospheric conditions. That should be discussed in the manuscript. Is there any idea regarding the sCI isomer/conformer concentrations in the experiments?

5) The reaction scheme in figure 4 should show how the two 1st generation $C_{10}$ closed-shell products from α-phellandrene ozonolysis as given in figures 2 and 3 are formed. Generally, some simplified reaction pathways in the given schemes are hard to understand. Please be more precise with the reaction equations, they should be "equations".

---

## Author Comment (AC1) · 12 Jan 2018

**Response to referees for, "Ozonolysis of α-phellandrene – Part 2: Compositional analysis of secondary organic aerosol highlights the role of stabilised Criegee intermediates" by Mackenzie-Rae et al.**

The authors would like to thank the referees for their time reviewing the manuscript, and for the thoughtful feedback provided. Based on their recommendations a number of modifications have been made to the original manuscript, which we believe has substantially improved the interpretation of this study. Presented below are the specific comments made by the reviewers (italicised) and our corresponding responses (non-italicised).

**Referee #1**

*The authors describe experimental findings of aerosol analysis from the gas-phase reaction of ozone with α-phellandrene based on filter measurement. Analysis has been carried out by a HPLC-MS technique using an Orbitrap tandem mass spec. Reaction conditions were chosen in such a way that secondary chemistry influenced the product distribution. Reactant concentrations are in most cases clearly higher than atmospheric levels. Addition of formic acid, acting as sCI scavenger, repressed bimolecular and most likely unimolecular sCI pathways resulting in changing product distribution. According to that, the authors concluded that sCI reactions are important for the aerosol constituents detected.*

*From my perspective there are some points that should be considered before acceptance can be suggested.*

> We thank the referee for their time reviewing this manuscript. We will now address their specific points.

*1) The authors used filter measurements. Nothing is said regarding possible aerosol processing between sampling and analysis. What was the delay time between sampling and analysis?*

> The filter samples were collected during smog chamber experiments conducted at the GIG-CAS in China, and analysed at the University of York, UK. As stated in the paper filter samples were, '...wrapped in foil and stored at 4°C prior to analysis' (p.3 l.25). Prior to storage the samples were dried in a desiccator for 24 hours, with this information now added. Samples were transported from China to the UK by an international courier, in an insulated box surrounded by dry ice, taking four days to deliver. Given that the transport temperature was lower than the storage temperatures used, and the short travel time, it is thought processing would have minimal impact on the obtained results. At the University of York samples were stored in a freezer at –20°C.

> Samples were stored for up to 1.5 years before analysis. Naturally the more volatile species would evaporate at a faster rate, although with respect to major peaks, little difference was observed between chromatograms taken in similar experiments one year apart. The more recent filter samples did have more minor peaks (i.e. a bumpier baseline), although analysis of these

was not the focus of this paper. So with regards to this study, it appears that storage time had little impact on results.

*2) What is the reason for the low RH < 5% in all runs?*

The impact of humidity on the reaction mechanism and subsequent SOA production from the ozonolysis of α-phellandrene was out of the scope of the current project. Therefore to keep experiments consistent with one another, similar humidity's were used throughout. Dry conditions were selected as it is easier to replicate experiments with low RH, all prior chamber characterisation (auxiliary mechanism) experiments had been performed under dry conditions, scavenging SCI (formic acid or $NO_2$) has less competition under drier conditions, and it can be assumed that the aerosol is free of any water phase (Cocker et al., 2001; Roldin et al., 2014).

Nonetheless the presence of water would change the competitive dynamics of SCI decomposition and makeup of the SOA. The effect of water on the ozonolysis of α-phellandrene thus would be of interest and is definitely an avenue of potential future research. Such studies have been performed on other monoterpenes with interesting results (e.g. Bonn et al. 2002; Cocker et al., 2001). Indeed a requirement for a comprehensive study into the effect of RH on SOA formation from α-phellandrene is a well studied and characterised dry state, which the current manuscript details.

Bonn et al. (2002) *J. Phys. Chem. A,* **106**, 2869.
Cocker et al. (2001) *Atmos. Environ.,* **35**, 6049.
Roldin et al. (2014) *Atmos. Chem. Phys.,* **14**, 7953.

*3) Unfortunately, there is no pair of experiments with formic acid ON or OFF and otherwise constant reaction conditions. The best pair is Exp.1 and Exp.7 where merely ozone was changed by a factor of about 2. Please show for these experiments the corresponding spectra, especially in the "dimer" range, which should confirm author´s conclusion of the importance of bimolecular sCI reactions for SOA formation. Identified "dimers" as given in table 4 should be marked in the spectra. Please state in the figure caption of all given spectra the EXP. number according to table 1.*

We agree with the referee that this would be a nice addition and we did consider this prior to submission. In many studies, the sample is directly injected into the ion source to produce a similar mass spectrum. We did not use this approach, as the high concentrations of monomers can cause the formation of clusters in the source that can be misinterpreted as dimers, thus producing an inaccurate picture of the SOA composition. In our study, the mass spectra presented in Figure 7 is an average mass spectrum taken across the chromatogram. For high concentration samples, such as in figure 7, the dimer region is clear. However, for low concentration samples, such as exp 1 and 7, the dimer region is dominated by contaminants from the water and methanol mobile phase and so the difference between samples is hidden within the noise. Therefore we do not feel that these spectra should be included.

Relevant experiment numbers have now been provided in captions for all spectra.

*4) Experiments have been conducted with higher reactant concentrations compared with atmospheric conditions resulting in higher steady-state sCI (and other product) concentrations as those expected in the atmosphere. Consequently, bimolecular pathways are less important under atmospheric conditions. That should be discussed in the manuscript. Is there any idea regarding the sCI isomer/conformer concentrations in the experiments?*

Indeed, the chamber conditions are higher than concentrations expected in the ambient atmosphere. The main change then is not the occurrence of bimolecular reactions, but in the type of bimolecular reaction occurring. In the ambient atmosphere, SCIs are predominantly consumed by $H_2O$ and possibly $NO_2$ in more urban areas. However, given the dry, $NO_x$ free conditions inside the reactor during the majority of these experiments, SCI-organic (e.g. aldehyde, ketones, acids, alcohols) or SCI-SCI/$RO_2$ accretion reactions are likely to be overrepresented with respect to the ambient. Whether these various bimolecular pathways occur fast enough to compete with the water concentration present inside the reactor is not entirely known, however results suggest that accretion reactions, in particular SCI-SCI reactions, are important. Using kinetics, if one assumes an RH of 5% and SCI reaction rate with water of $1\times10^{-16}$ cm$^3$ s$^{-1}$ (Welz et al., 2014), the bimolecular reaction rate of SCIs with water is calculated as 3.85 molecules s$^{-1}$. Experiment 10 had 175 ppb of a-phellandrene, so taking 175 ppb as an upper SCI/$RO_2$/organic concentration limit, a rate constant larger than $8.9\times10^{-13}$ cm$^3$ molecule$^{-1}$ s$^{-1}$ is required for other bimolecular pathways to outcompete the SCI reaction with water inside the reactor. Theoretical and experimental measurements have shown this is possible for simple CIs (e.g. Vereecken et al., 2015), however research is limited, and how rates extend to larger, more substituted α-phellandrene derived SCIs is unknown.

If these SCI-organic bimolecular reactions were prominent inside the reactor it would result in enhanced dimer production. Increased dimerisation would lead to more SOA, which is already amplified by higher precursor concentrations. Elevated peroxy radical cross-reactions (representative of a $NO_x$ free environment) also impacts the product distribution, with the ramification of this on SOA discussed in the companion paper. Artificially enhanced SOA loadings due to these various reasons can then lead to enhanced partitioning of less volatile species into the organic mass (mentioned on page 8, line 20).

Discussion of the impact of higher precursor concentrations is now included through addition of the following to the manuscript on page 12 line 7:

"One important caveat is that smog chamber conditions are not entirely representative of the ambient, with these experiments conducted using artificially high precursor concentrations. Whereas in the atmosphere SCI are largely scavenged by water or water dimers (Vereecken et al.,

2015), the dry conditions and increased availability of organics inside the reactor increases propensity of the various SCI accretion channels."

Theoretical work has shown both double bonds in α-phellandrene to be similarly reactive, with all four initial CI conformers possible. Functional groups can equally be formed from various reaction pathways e.g. a carboxylic acid can form from unimolecular rearrangement through a dioxirane or secondary ozonide, or from a bimolecular reaction of an SCI with water; carbonyls can be formed directly from addition of ozone, or through bimolecular reactions etc. Therefore it is only products formed through the hydroperoxide (HP) channel that give an indication into initial SCI conformation. The hydroperoxide channel is possible for excited CIs however, and so may not give a true indication of SCI formation for each of the different conformers. Furthermore, the HP channel is predicted to be involved in a small number of proposed products, and if inferences were to be drawn from these one would need to assume equal extraction and ionisability of the different products. For these reasons, the present study is unable to provide any real indication of relative isomeric concentrations of SCI from a-phellandrene. Nonetheless elucidated pathways to identified products are derived from a number of different initial CI conformers, supporting theoretical work that all CI conformers contribute.

*5) The reaction scheme in figure 4 should show how the two $1^{st}$ generation $C_{10}$ closed-shell products from α-phellandrene ozonolysis as given in figures 2 and 3 are formed. Generally, some simplified reaction pathways in the given schemes are hard to understand. Please be more precise with the reaction equations, they should be "equations".*

The first step in Figure 3 has been clarified, as requested by the second referee. Additional detail for certain reaction steps in Figure 4 is now given. Clarification of implicit reactions is additionally provided in the caption.

**Referee #2**

*The study by Mackenzie-Rae et al. presents interesting new results on the chemical composition of SOA from ozonolysis of alpha-phellandrene. In general the results are well presented and discussed, though it is somewhat disappointing that a manuscript authored by several native English-speaking persons contains a number of grammatical errors and unclear sentences. I recommend that the work is published after the following points have been adequately addressed.*

We appreciate the referee's feedback and their recognition of the value that the study offers the scientific community. We will now address their specific concerns to improve the precision, clarity and discussion of the manuscript.

*Page 1 Abstract: The abstract must mention that the study only investigates water-extractable components of the SOA.*

We have included this additional wording.

*Page 1 Line 16: Why is quadrupole written with capital first letter? Since it is not used for an abbreviation, nor is it a name, lower case would be more correct.*

This has been amended.

*Page 1 Line 29: "around Eucalypt forests regions" -> please correct the grammar here*

Has been changed to, 'around eucalypt forests'.

*Page 2 Line 13: "One monoterpene for which little study has been conducted is α-phellandrene." -> please rewrite to make the sentence more clear and correct*

Has been amended to, 'Despite this, the literature has predominantly focused on a small number of the more commonly emitted monoterpenes (e.g. α-pinene, β-pinene, limonene), which, whilst important on the global scale, fails to consider the impact that other monoterpenes may be having on local environments. For instance, α-phellandrene is one of the most reactive monoterpenes (Atkinson and Arey, 2003), yet relatively little is known about its tropospheric degradation and subsequent aerosol forming characteristics'.

*Page 2 Line 17: "Believed then to be of particular importance is the abundance of α-phellandrene in extracts of numerous species of Eucalypts" -> please rewrite to make the sentence more clear*

Sentence structuring has been modified to improve clarity: 'α-phellandrene has been identified in high abundance in the extracts of numerous species of eucalypts (Brophy and Southwell, 2002; Li et al., 1995; Maghsoodlou et al., 2015; Pavlova et al., 2015), with monoterpene emission rates known to increase linearly with concentration in plant tissue (Fuentes et al., 2000). Indeed Maleknia et al. (2009) and He et al. (2000) have both identified α-phellandrene in emissions from various eucalypt species in the laboratory, therefore α-phellandrene is likely a non-negligible contributor to the large monoterpene emissions reported from eucalypt forests (Emmerson et al., 2016).'

*Page 2 Line 28: "However, postulated gas-phase species could not explain the properties of the SOA observed." -> please make it clear that you refer to the companion paper here.*

Has been changed to, 'However, gas-phase species postulated in the companion paper could not explain the properties of the SOA observed.'

*Page 3 Line 6: Please also refer to some of the first studies that applied this technique to aerosols.*

We have added two additional references here.

Gao, S., Keywood, M., Ng, N. L.;, Surratt, J. D., Varutbangkul, V., Bahreini, R., Flagan, R. C., Seinfeld, J. H.,  Low-Molecular-Weight and Oligomeric Components in Secondary Organic Aerosol from the Ozonolysis of Cycloalkenes and α-Pinene.  J. Phys. Chem. A, 108, 10147 2004.

Warnke, J., Bandur, R., Hoffmann, T. Capillary-HPLC-ESI-MS/MS method for the determination of acidic products from the oxidation of monoterpenes in atmospheric aerosol samples.  Anal. Bioanal. Chem., 385, 34-45, 2006.

*Page 3 Line 8-9:  I suggest to remove "and relevant to the current discussion"*

  Thanks, the text has been removed.

*Page 3 Line 25-30: The storage temperature is higher than often used in other studies (-18C). For how long were the samples stored and can you comment on the possible influence on the analytical results?*

  The samples were stored at both 4°C (fridge) and -20°C (freezer) before analysis. To be safe only the maximum temperature was quoted in the paper. See response to referee 1 above.

*Page 3 Line 29: Why were the filters only extracted in water, not a polar solvent? It is important that this is emphasized throughout the manuscript - unless the authors have results showing that most of the SOA is actually extracted by this method. If so, those results or previous studies should be mentioned.*

  The authors have previously tested the efficiency of different solvents for SOA.  Water extracts the majority of the SOA and is comparable to studies of ambient aerosol, which look at the WSOC.  Water gives a very clean extraction with little impact on the efficiency and so is considered the optimum solvent. A reference has been added.

  Hamilton, J.F., Lewis, A.C., Carey, T.J., Wenger, J.C. Characterization of Polar Compounds and Oligomers in Secondary Organic Aerosol Using Liquid Chromatography Coupled to Mass Spectrometry, *Analytical Chemistry,* 80, 474-480,  2008**.**

*Page 4 Line 1: Was the effect of "evaporation to dryness" on molecular composition of samples investigated?*

  This was not investigated in this study.  The vacuum solvent evaporator uses high vacuum and low temperature (<20 C) to remove the water. Therefore evaporative losses should be minimized.

*Page 4 line 6: "reverse-phase" should be corrected to "reversed-phase".*

  This been corrected.

*Page 4 line 9: Did you actually inject 30 mL? Please provide dimensions of the column and eluent flow rate.*

This is a typo and should say 30 μl. This has been corrected.

*Page 4 line 15: How frequently was the mass calibration done? Please be specific.*

The mass calibration was done every day.  This has been added.

*Page 4 Line 19-20: It seems strange that you mention methylation during extraction, when you extracted in water. The influence of adduct formation with methanol was further investigated by Bateman et al. (2008) and Kristensen and Glasius (2011), also showing that methanol esters can form during HPLC analysis.*

The SOA ample is extracted into water first, then evaporated and reconstituted in 50:50 methanol water. We have found this improves the chromatography without producing methanol esters.

*Page 4 Line 23: "Because all first- and second-generation products of α-phellandrene ozonolysis contain at least one functional group that is capable of ionisation (Mackenzie-Rae et al., 2016, 2017a), it is reasonable to assume that a high proportion of water soluble SOA components will be observed.". Especially the first part of the sentence is quite a bold statement, which should be modified. The next sentences on ionization must be shortened to avoid repetition.*

As it stands the statement initial part of the sentence is quite bold. This is because a large fraction of products contain at least one carbonyl group and so are expected to be detected in the positive mode. Nonetheless, there does exist the possibility of species forming that do not contain readily ionisable groups, with the sentence now modified to reflect this: 'Because a large fraction of first- and second-generation products…'.

The line, 'with the analyte R being observed as [R–H]$^-$ ion in the negative mode and [R+Na]$^+$ ion in the positive mode' was deleted to avoid the repetition.

*Page 5 Line 19: "mass spectra are" or "mass spectrum is". Please correct.*

Thank-you, has been corrected to, 'mass spectrum is'.

*Page 7 Line 9: Can you add a reference where the loss of 88 Da is observed from diacids?*

We have included a reference for this from the literature.  However, the main reason for this statement is based on our previous extensive analysis of standard compounds and other SOA components.

*Page 7 Line 18: Since the vapour pressures have not been measured, please change to "estimated low vapour pressures".*

Has been changed.

*Page 7 Line 30: Has this process not been investigated since it was suggested in 2008?*

The cited 2008 study by Walser et al. claim that oxidation of aldehydes by ozone is faster in the aerosol phase. Thus it is possible that condensed-phase aldehydes are oxidized by ozone to carboxylic acids. The authors cannot find other studies that make this similar claim, although there exists plenty of literature concerning ozone involvement in heterogeneous reactions. To clarify that it is the heterogeneous reaction of ozone causing the oxidation, the following is added to p. 7 l. 31: '...through heterogeneous reaction with ozone'.

*Page 8 Line 21: "it is not unreasonable to assume enhanced partitioning of these prominent intermediate-volatile gas-phase species". What do you mean by enhanced? Compared to what?*

The statement was intended to convey the notion that, despite having high estimated vapour pressures, which alone would almost exclude these species from any meaningful SOA contribution, due to the aforementioned reasons it is indeed possible that they are significant contributors. Enhanced then was the wrong word, and has been changed to 'considerable'.

*Page 8 Line 30-31: The last sentence is quite unclear.*

Has been broken into two sentences and re-written to improve clarity: "As shown in Fig. 6, the estimated saturation vapour concentrations of P7 and P8 are not sufficient for nucleation. Nonetheless their presence in the filter samples suggests autoxidation is occurring during α-phellandrene oxidation, with it entirely possible that, upon further intramolecular H-transfers, volatilities will be sufficiently reduced to form the extremely low volatility compounds required for nucleation (Ehn et al., 2014; Jokinen et al., 2015)."

*Page 9 Line 2-3: "Whilst the SOA analysed consists of a complex mixture of compounds, chromatographic analysis suggests that the SOA is dominated by a small number of major constituents." Please write more clearly what you mean - is it a complex mixture or a small number of major constituents?*

Has been changed to, "Whilst it was anticipated that the SOA would consist of a complex mixture of compounds, chromatographic analysis suggests that the SOA is instead dominated by a small number of major constituents".

*Page 9 Line 5: Please mention the tool you used for this, since these estimates vary considerably.*

The estimative tools have now been provided in-text. The supplementary entry describing the estimative methods in detail has been retained.

*Page 9 Line 15-16: Please clarify this sentence.*

Has been re-written: "Three products were determined to be of low volatility, being $C_{10}$ species with between 5 and 7 oxygen atoms. Both

first- and second-generation species are classified as SVOCs."

*Page 10 Line 25-26: Please order these references by publication year.*

All in-text references throughout the paper have been ordered alphabetically, as per ACP instructions, 'In terms of in-text citations, the order can be based on relevance, as well as chronological or alphabetical listing, depending on the author's preference.' This is the case for these references, and is kept the same to remaining consistent with the remaining references in the manuscript.

*Page 15 Line 6: Please add information on the reaction rate with water.*

Has been included.

*Additional reference: The study of Zhang et al. (2015) provided new information on molecular composition and dimers in monoterpene SOA and is suggested as an additional reference.*

The Zhang et al. (2015) paper did provide new, interesting temporal insights into SOA composition from α-pinene ozonolysis. Reference to it has now been made.

*Figure 3: The first step needs additional information.*

Has been provided.

*Table 1: It should be clear from the table caption, that most of this information is from the companion paper.*

Reference to the companion paper is now given in the caption.

*References: A.P. Bateman et al., Environmental Science and Technology (2008) 42, 7341-7346 K. Kristensen and M. Glasius, Atmospheric Environment (2011) 45, 4546- 4556. Y. Zhang et al., Proc. Natl. Acad. Sci. (2015) 112, 14168-14173.*

---

## Author Response (AR2)

Dear Frank

Thank you for your final editor comments. I have outlined our response below in italics below. I have also added the doi number for the mass spectra data archive.

*Page 4: This sentence has been changed to make it clearer.*

*Previous studies have shown that water can extract the majority of the SOA mass formed in chamber experiments, with a very low background organic signature (Hamilton et al., 2008).*

I would like you to consider adding two comments, with respect to the quantitative interpretation of the observed signals.

(i) It would be useful to have a comment how quantitatively the peak intensities are. In other words do you expect any sensitivity difference for different compound types. This does not affect the actual message of your work, but it would be useful for readers not as familiar with ESI and soft ionization techniques.

*We have added some explanatory text to convey this message on page 5.*

*Ionisation efficiency of individual molecules can vary significantly in ESI and so the largest peak may not represent the most abundant species.*

(ii) Regarding the point you raise for product P1 m/z151. This has a very high vapor pressure and I think the possibility of some decomposition during analysis has to be considered, e.g. some reversible oligomer. I do not disagree that there could be some partitioning but it is surprising that this is the biggest peak in the positive mode, by far. Would you consider adding a sentence in the conclusion that addresses this point, for example, that further work investigation the origin of the high vapor pressure compounds that result in large signals?

*We have added that this signal may result from a reverse oligomerisation process during the analysis.*

*On page 8:*
*In addition, they may be present as a result of decomposition of larger species during the analysis (e.g. reverse oligomerisation).*

*In conclusions:*
*Further investigation is required however to better characterise the role of stabilised Criegee intermediates in the system, to parameterise the specific accretion mechanisms occurring and to determine the origin of the high vapour pressure compounds that correspond to the large positive mode signals.*